# Effects of bacterial composition and aquatic habitat metabolites on malaria vector larval availability in irrigated and non-irrigated sites of Homa Bay county, western Kenya

Pauline Winnie Orondo[1]*, Kevin O. Ochwedo[2], Harrysone Atieli[2], Guiyun Yan[3], Andrew K. Githeko[4], Steven G. Nyanjom[1]*

1 Department of Biochemistry, Jomo Kenyatta University of Agriculture and Technology, Nairobi, Kenya, 2 International Center of Excellence for Malaria Research, Tom Mboya University, College of Maseno University, Homa Bay, Kenya, 3 Program in Public Health, College of Health Sciences, University of California at Irvine, Irvine, California, United States of America, 4 Centre for Global Health Research, Kenya Medical Research Institute, Kisumu, Kenya

* paulineorondo@gmail.com (PWO); snyanjom@jkuat.ac.ke (SGN)

## Abstract

Gravid *Anopheles* malaria vectors depend on both chemical and physical (including microbial) cues for selection of preferred habitats for oviposition. This study focused on assessing the effects of bacterial composition and habitat metabolites on malaria vector larval availability in irrigated and non-irrigated potential larval sources. Water samples were collected from larval positive and negative habitats in the irrigated and non-irrigated areas of Homa Bay county. Bacteria cultured from the water samples were subjected to Matrix Assisted Laser Desorption Ionization-Time of Flight Mass Spectrometry (MALDI-TOF MS) for species identification. DNA was extracted from the colonies and polymerase chain reaction (PCR) and sequencing done. Finally, the metabolite composition of larval positive and negative habitats was determined. MALDI-TOF MS results revealed that *Bacillus* was the only genera identified from larval sources in the non-irrigated zone. In the irrigated area, *Shigella* was the dominant genera (47%) while *Escherichia coli* was the abundant species (13/51). Of the sequenced isolates, 65% were *Bacillus*. Larvicidal isolates *Brevibacillus brevis*, *Bacillus subtilis*, and *Exiguobacterium profundum* were isolated and grouped with *Bacillus mojavensis*, *Bacillus tequilensis*, *Bacillus stercoris*, and *Brevibacillus agri*. Irrigated areas with larvae had reduced crude fat (0.01%) and protein content (0.13%) in comparison to those without larvae. In irrigated and non-irrigated areas, larval presence was evident in habitats with high total chlorophyll content (1.12 μg/g vs 0.81μg/g and 3.37 μg/g vs 0.82). Aquatic habitats with larvae in both irrigated and non-irrigated areas exhibited higher sugar concentration than habitats without larvae; however, when compared, non-irrigated areas with larvae had higher sugar concentration than similar habitats in irrigated areas. In addition, substantial concentrations of Manganese, Calcium, and Copper were found in aquatic habitats containing larvae in both irrigated and non-irrigated areas. These results allow for prospective examination as potential larvicidal or

**Data Availability Statement:** All relevant data are within the paper and its Supporting information files.

**Funding:** The work was supported by grants from the International Foundation for Science (https://www.ifs.se/) grant number I2-A-6605-1 and The Mawazo Institute Research Funds (https://mawazoinstitute.org/) grant numbers 2022-0-07 and 2022-3-09 awarded to PWO. These grants supported the whole research project including data collection, analysis and publication.

**Competing interests:** The authors have declared that no competing interests exist.

adulticidal agents and could be considered when designing potential vector control interventions.

## Introduction

Malaria is a parasitic infection that results from an infective bite by the *Anopheles* mosquito. These vectors are widely distributed within the tropics of Africa [1], Asia [2–4], and Central America [5]. In Kenya, the main vectors have been observed to be complexes of *An. gambiae* and *An. funestus* which occur in sympatry over vast geographical regions [6, 7]. The distribution of these malaria vectors has been greatly linked to the aquatic habitat composition as the different species prefer specific habitats. In addition to the physical characteristics of the habitats, there are known to be other chemical and biological cues that attract for oviposition of gravid mosquitoes [8]. These oviposition sites have differing composition of both biotic and abiotic composition [9]. The initial stages of a mosquito's lifecycle is fully aquatic and this stage significantly affects their adult life in terms of survival, fitness, and reproduction [10]. The *Anopheles* mosquito has been observed to select for habitats that favor the survival and development of the immature [11].

Several physical and chemical factors influence the choice of oviposition sites for malaria vectors. Previous studies have shown that gravid malaria mosquitoes are attracted by water vapor [12] and olfactory cues from con-specific larvae [13, 14], specific microbiome [13, 15], and soil composition [16] present in the aquatic habitat. Microbes [8, 17–19], larval presence [8, 20], decomposing debris that serves as food for the larvae [21], and predators in natural habitats [8] are all known sources of chemical cues. Additionally, gravid mosquitoes are also attracted by several chemical compounds for oviposition while others deter or have no effect on oviposition [8].

Microbial volatiles from larval habitats influence oviposition by mosquitoes [9, 22]. This is because micro-organisms act as food for the larvae. Studies have shown that algae may enhance *An gambiae* selection for habitat for oviposition [23]. As these bacterial volatiles act as attractants to different species [8], reduced oviposition has been observed with a reduction in bacterial concentration in aquatic habitats [24, 25]. Further studies are needed to confirm the oviposition attractant compounds for the development of mosquito traps and/ or in combination with larvicides for vector control. Additionally, volatiles produced by organic matter have been observed to attract gravid female *Anopheles* species for oviposition [26].

Preliminary studies indicate that aquatic habitat cues may attract mosquitoes from a greater distance. Aquatic vegetation may produce cues, as emergent vegetation frequently arises from and surround malaria mosquito larval sites. Different vectors were observed to prefer certain habitat types to others, however, this has been observed to be changing over time. Originally it was understood that *An. funestus* bred mainly in more permanent marshes and swamps with tall vegetation [9, 27, 28]. This notion is however changing as this species is now observed to inhabit habitats without vegetation [29]. *An. gambiae* s.l. has been thought to prefer the edges of fresh slow-moving temporary waters which are sunlit [9], however, recent studies have shown that these vectors can also infect more permanent habitats with vegetation cover like rice paddies [30]. Vegetation also produces specific volatile compounds that can act as attractants or repellants. Attractants that have been isolated include a mixture of terpenoid and alcohol compounds [31].

Therefore, it is crucial to understand the characteristics of oviposition sites as it is a prospective are if exploration towards the development of effective vector management strategies for insect-borne infectious diseases [32]. Knowledge of the determinants that can be used as

oviposition repellants or attractants is essential in this process. Therefore, this study sought to understand the chemical cues in aquatic larval mosquito habitats that attract or repel gravid female malaria vectors.

## Materials and methods

### Study site

The study was conducted in Homa Bay County, western Kenya. Homa Bay County is a semi-arid area located along the shores of Lake Victoria (34.6˚E and 0.5˚S; 1,330 m above sea level) (Fig 1). The area has been previously described [30, 33] and is identified as malaria endemic region [34]. Briefly, the study site is composed of irrigated and non-irrigated areas. The irrigated area is composed of a channel-based irrigation system that assists in farming at the household level while the non-irrigated area has no irrigation channels and relies on rainfall to sustain farming and agricultural activities. The agricultural practices in this area are mainly of the subsistence type with a little proportion of farmers practicing cash crop farming of cotton. The main crops grown in this area includes maize, millet, beans, groundnuts, fruits, rice, and cotton while the major livestock kept in this area include cattle, goats, sheep, and donkeys. Fish farming is mainly practiced along the lake shores and in the irrigated areas.

### Sample collection

Water samples were collected between November 2021 to January 2022 from the irrigated and non-irrigated areas from larval-infested and non-infested aquatic habitats. Larval-positive habitats were identified as those that harbored malaria vectors while larval-negative habitats had no malaria vectors during the sampling time. The water samples were pooled into 12 bottle sets and used for microbial analysis. From each aquatic habitat, a minimum of 50 ml of water was collected. Pooling occurred monthly with a minimum volume of 1 liter per pool (Table 1). The water was transported to the laboratory and stored at -20 ˚C awaiting analysis.

For malaria vector presence confirmation, larval sampling using the standard dipping technique with the help of a standard dipper of 350 ml [35] was used. During sampling, Anopheline larvae were morphologically identified using [1, 36–38] protocols.

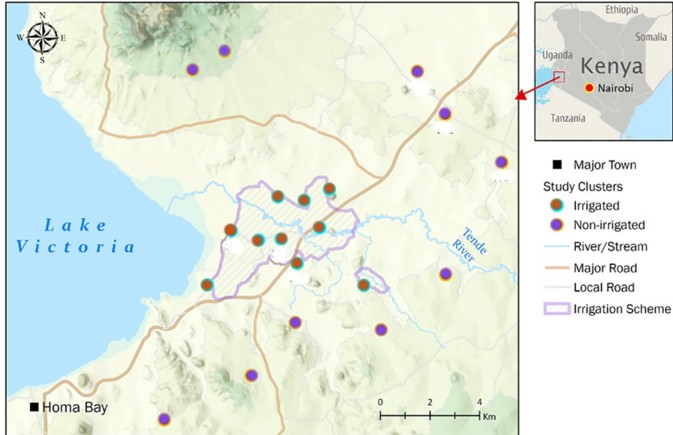

**Fig 1. Map of the Homa Bay study site showing the sampling areas (colored) in the irrigated and non-irrigated areas.**

**Table 1. Survey period, targeted areas for larval sources, and larval availability.**

| Month/ Area | Irrigated | | Non-irrigated | |
|---|---|---|---|---|
| November 2021 | Larval positive | Larval negative | Larval positive | Larval negative |
| December 2021 | Larval positive | Larval negative | Larval positive | Larval negative |
| January 2022 | Larval positive | Larval negative | Larval positive | Larval negative |

## Bacterial analysis

**Bacterial culture.** Peptone water was added and incubated at 36˚C ± 2˚C for 16 to 20 hours. Bacteria was cultured in nutrient agar at 37˚C for 24 hours and single colonies were plated on nutrient agar, Eosin Methylene Blue (EMB) agar, and MacConkey agar at 37˚C for 24 hours. Individual colonies were then grown in nutrient broth for 24 hours and preserved in 50% glycerol for Matrix-Assisted Laser Desorption Ionization-Time of Flight Mass Spectrometry (MALDI-TOF MS) and sequencing.

## Microbial culture analysis by MALDI-TOF MS

Each bacterial colony identified from the culture was placed in duplicate on the MALDI-TOF plate target. The direct smear plus formic acid sample preparation method was utilized. Briefly, a colony was transferred from the agar surface to the target plate and was overlaid with 0.5 μl of 25% formic acid solution. After drying, 0.5 ul of a saturated solution of alpha-cyano-4-hydroxycinnamic acid (10mg/ml) diluted in 250 ul solutions comprising of acetonitrile, HPLC grade water, and ethanol in a ratio of 3:3:3 and containing a final concentration of 3% of trifluoroacetic acid was added. The target plate was then analyzed using the MALDI-TOF spectrometry for bacterial identification against the SARAMIS database. Spectra range recording was done in linear mode with the mass ranging between 2000 to 20000 Daltons. Genotyping of the colonies was done by MALDI-TOF MS and the plate was then read using the Shimadzu software (Axima Confidence, Shimadzu, Japan).

## Polymerase chain reaction and sequencing

Bacterial DNA was extracted from 31 selected colonies using the ISOLATE II Genomic DNA Kit (Bioline, London, UK) following the manufacturer's instructions. Polymerase chain reaction (PCR) was done using Go Taq Master Mix (Promega, Wisconsin, USA). A final PCR reaction volume of 25 μl was prepared by adding 12.5 μl of Go Taq master mix, 9.5μl nuclease-free water, 0.3 μl each of the forward and reverse primers, and 2 μl of the bacterial DNA template. The PCR reaction was set in an ABI GeneAMP 9700 (Applied Biosystem, Foster City, CA, USA) as follows: 95˚C for 5min, 35 cycles (94˚C for 30 sec, 57˚C for 45 sec, 72˚C for 45 sec), followed by a final extension for 7 mins at 72˚C. The quality of the amplicons was done using gel electrophoresis in 1.5% w/v agarose gel before purification and sequencing. Each of the cleaned amplicons (using Exonuclease I and Shrimp Alkaline Phosphatase or ExoSAP-IT) were bi-directionally sequenced using BigDye® Terminator v3.1 Sequencing Standard kit on ABI PRISM® 3700 DNA Analyzer (Applied Biosystems, Foster City, CA, USA).

## Metabolite identification in irrigated and non-irrigated larval habitats

The contents of crude protein, polyphenols, free radicals, fatty acids, fats, and other minerals were determined according to Association of Official Analytical Chemists (AOAC)® protocols [39]. These protocols were modified as illustrated by Wanjiku EK [40].

## Crude protein method

The extraction followed Wanjiku EK [40] protocol. 10 ml of the sample was mixed with the catalyst (5g $K_2SO_4$, 0.5g $CuSO_4$, and 15 ml concentrated $H_2SO_4$). The resulting mixture was heated to a blue color, cooled, then transferred into a 100 ml volumetric flask which was topped up with distilled water. A blank digestion was additionally carried out using the catalysts and acid. 10 ml of the diluted digest was then transferred into a distilling flask andwashed with 2 ml of distilled water. Finally, the distillate was titrated to an orange color with 0.02N − HCl.

## Determination of total polyphenols

Determination of total polyphenol was carried out following the method of Waterman [41], albeit with slight modification [40]. 10 ml of the sample was put in an amber glass bottle and 50 ml of methanol was added. Extraction was done for 3 hours in a shaker and the extract kept in the dark for 72 hours, filtered, then topped up to 50 ml using methanol. The extract was then centrifuged for 10 min at 25°C at 150 rpm. Thereafter, 2 ml of the supernatant was filtered using 0.45 μl micro-filter into a test tube. 2 ml of 10% Folin-Ciocateu was then added, vortexed, and 4 ml of 0.7M sodium carbonate added and vortexed again. The extract was incubated for two hours to develop color and absorbance was read at 765 nm using gallic acid as a standard in a UV-Vis spectrophotometer (Shimadzu model UV– 1601 PC, Kyoto, Japan).

## Fatty acid analysis

Gas chromatography (GC) was used to determine the fatty acid profile. The Bligh and Dyer [42] method was modified for extraction [40]. 100 ml of the sample was mixed with 50 ml of hexane and the mixture shaken overnight. 2 mg of the oil sample was refluxed in 5 ml of 95% methanol-HCl for 1 hour. Three portions of 5 ml hexane were used to extract the methyl esters and then washed with 5 ml of distilled water. A vacuum rotary evaporator was used to dry the hexane layer, and the residue re-dissolved in 1 ml of hexane. Afterward, 1 μl was injected into the GC, under split mode of 60 (Shimadzu GC-2010 equipped with auto-sampler) using a supelcowax capillary column of 30 m x 0.53 mm, at an injection temperature of 240°C and a detection temperature of 260°C for under a flame ionization detector. Comparing the retention periods against the standards and expressing the results as a percentage of all methyl esters allowed for the identification of fatty acid methyl esters.

## Determination of the free radical scavenging activity

A UV spectrophotometer at 517 nm was used to test the radical scavenging activities of plant extracts against the 2, 2-Diphenyl-1-picryl hydrazyl (DPPH) radical (Sigma-Aldrich) as described by Molyneux [43]. The extracts were prepared in methanol (Analar grade) at the following concentrations: 0.01, 0.1, 1.0, 2.0, and 5 mg/ml. Vitamin C, the antioxidant standard, was used at the same concentrations of the extracts. In a test tube, 1 ml of the extract was put followed by 3 ml of methanol and 0.5 ml of 1 mM DPPH in methanol. The same amount of methanol and DPPH were used to prepare a blank solution. A UV-Vis spectrophotometer (Shimadzu model UV—1601 PC, Kyoto, Japan) was zeroed with methanol, and the absorbance was read at 517 nm after 5 minutes.

## Mineral analysis

Following the AOAC® protocol [44], 100 ml of the water sample was added in a 250 ml beaker and placed on a heating plate. The water was heated and while about to boil, 2 ml of nitric

acid-water mixture (50:50) and 10 ml HCl-water mixture (50:50) were added. The mixture was decreased to around 25 ml by evaporating the samples. The 25 ml residues were transferred into 100 ml volumetric flasks, which were then filled with distilled water to the mark. A Shimadzu Atomic Absorption Spectrophotometer, Model AA-7000, was used to determine the lead, cadmium, mercury, and arsenic content of the samples.

## Lipid/ oil content analysis

Gas chromatography was used to determine the fatty acid profile. A modified version of the Bligh and Dyer [42] protocol was applied for lipid extraction [40]. Samples of 100 ml were mixed with 50 ml of hexane and the mixture shaken for 30 minutes and then left to stand. After collection of the hexane layer and the aqueous layer returned. The extraction was repeated once again. The hexane fractions were combined and filtered using defatted cotton wool and anhydrous $Na_2SO_4$ to remove the water. The filtrate was concentrated at 40˚C and dried for 30 minutes in an oven at 70˚C.

## Determination of glucose, pectin, amylose, and cellulose content

This procedure followed AOAC [45] protocol. 50 ml acetonitrile and 50 ml of water were mixed. 2 ml of lead acetate was added. The solution was filtered in 5% anhydrous oxalate and micro-filtered. A high-performance liquid chromatography (HPLC) system (Model LC-20A Series, Shimadzu Corp., Kyoto, Japan) fitted with a refractive index (RI) detector was used for analyzing individual sugars. The flow rate was 0.5–1.0 ml/min, the injection volume was 20 μl, and the oven temperature was set to 30˚C. Acetonitrile and water in a volume ratio of 3:1 was used as the mobile phase. Using the standards, the sugars present were identified, and their individual concentrations calculated.

## Extraction of chlorophyll and carotenoid

This followed Arnon, [46] protocol. 25ml of the sample was added to 25ml of heated acetone. This mixture was centrifuged at 5000–10000 rpm for 5 mins. The supernatant was transferred to another tube and the procedure was repeated until the residue became colorless. The absorbance of the solution was read at 480nm, 645nm, and 663nm against the solvent (acetone) which was the blank.

## Statistical analysis

**MALDI-TOF MS.** To perform analyses on the MALDI-TOF MS spectra and the final character data set, various similarity measures were evaluated. Band peaks were used to calculate similarities with available data sets to produce comparable identification results.

**Sequence alignment and analysis.** *De novo* assembly of the respective 31 generated raw reads was done using Geneious Prime® version 2022.2.2. The generated nucleotide consensus sequences were blasted using Basic Local Alignment Search Tool (BLASTn) against GenBank database for comparison and identification. The accessions from GenBank with high similarity index were retrieved and alignment done using ClustalW algorithm within Mega version 11.0.13 software. The evolutionary relationship between the isolated and retrieved GenBank sequences was computed in Mega version 11.0.13 software [47].

**Crude protein method.** Calculations were done using the formulae;

$$Nitrogen\% = (V1 - V2) \times N \times F \times 0.014 \times 100/V \times 100/S$$

Where; V1 = Titre for the sample (ml);

V2-Titre for the blank (ml)

N = Normality of standard HCL solution

F = Factor of standard HCL solution

V = Volume of diluted digest taken for distillation (10ml)

S = Weight of the sample taken (g)

$$\text{Protein \%} = \text{Nitrogen} \times \text{protein factor}$$

**Determination of the free radical scavenging activity.** The radical scavenging activity was calculated using the following formula:

$$\text{\% inhibition of DPPH} = \{(A_B - A_A)/A_B\} \text{ x } 100$$

Where; $A_B$ is the absorption of the blank sample

$A_A$ is the absorption of the tested extract solution.

The results were expressed as percentage DPPH inhibition and mean inhibitory concentrations ($IC_{50}$) calculated from a plot of % DPPH inhibition against the concentration of extract.

**Fatty acid method.** Crude fat was calculated as follows:

$$\text{\% Crude fat} = \frac{W2 - W1(g)}{S(g)} \times 100$$

Where; Weight of empty flask (g) = W1,

Weight of flask and extracted fat (g) = W2,

Weight of sample = S

**Estimation of chlorophyll content.** Concentrations of chlorophyll a, chlorophyll b, total chlorophyll, and carotenoid were calculated using the following equations:

Total Chlorophyll : $20.2(A645) + 8.02(A663)$

Chlorophyll a : $12.7(A663) - 2.69(A645)$

Chlorophyll b : $22.9(A645) - 4.68(A663)$

Carotenoid : $[A480 + (0.114(A663) - (0.638 - A645)] \times V/1000 \times W$

## Results

### Bacterial isolates detected by MALDI-TOF MS from irrigated and non-irrigated larval sources

Nutrient agar showed growth for both irrigated and non-irrigated larval positive and negative water samples. A total of seven bacterial genera were identified using MALDI-TOF MS namely *Bacillus*, *Brevibacillus*, *Citrobacter*, *Enterobacter*, *Escherichia*, *Klebsiella*, and *Shigella*. From larval sources in the non-irrigated zone, only *Bacillus* (*Bacillus amyloliquefaciens*) was identified whereas the rest of the genera were from larval sources in irrigated zones. *Shigella* was the dominant (24/51, 47%) genera among the isolates followed by *Escherichia* (13/51, 25%) whereas *Bacillus* (4/51, 8%), *Citrobacter* (4/51, 8%), *Brevibacillus* (2/51, 4%), *Enterobacter* (2/51, 4%), and *Klebsiella* (2/51, 4%) were observed at a lower frequency (Table 2). *Escherichia coli* was the abundant species (13/51) followed by *Shigella sonnei* (11/51), *Shigella boydii* (8/51), *Shigella flexneri* (3/51), and *Bacillus amyloliquefaciens* (3/51). The rest *Citrobacter freundi*, *Citrobacter braaki*, and *Klebsiella pneumoniae* were each detected twice (Fig 2). Other bacteria species identified included *Enterobacter cloacae*, *Shigella sp.*, *Enterobacter asburiae*, *Shigella dysentriae*, *Bacillus subtilis*, *Brevibacillus agri*, and *Brevibacillus brevi* (Fig 2).

**Table 2. Distribution of bacterial genera detected by MALDI-TOF MS from irrigated and non-irrigated larval sources.**

| Genera | Irrigated area | | Non-irrigated area | |
| --- | --- | --- | --- | --- |
| | Larval positive habitats (n) | Larval negative habitats (n) | Larval positive habitats (n) | Larval negative habitats (n) |
| *Bacillus* | 1 | 2 | - | 1 |
| *Brevibacillus* | 2 | 0 | - | - |
| *Citrobacter* | 0 | 4 | - | - |
| *Enterobacter* | 2 | 0 | - | - |
| *Escherichia* | 7 | 6 | - | - |
| *Klebsiella* | 1 | 1 | - | - |
| *Shigella* | 10 | 14 | - | - |

**n**: Number of isolates

Key gram-positive bacteria species in larval present habitats in irrigated areas included *B. amyloliquefaciens*, *B. agri*, and *B. brevi* whereas gram-negative included *E. asburiae*, *E. cloacae*, *K. pneumoniae*, *S. boydii*, *S. dysentriae*, *S. flexneri*, and *S. sonnei*. In larval-negative habitats, gram-positive bacteria included *B. amyloliquefaciens*, and *B. subtilis*, whereas gram-negative bacteria were *C. braaki*, *C. freundi*, *E. coli*, *K. pneumoniae*, *S. boydii*, *S. flexneri*, *S. sonnei*, and *Shigella* spp. Larval sources in irrigated areas had high bacterial abundance and diversity (Fig 3). Habitats within irrigated area that lacked larvae had higher bacterial species abundance as compared to habitats with larvae (27 against 23). Both habitat types however had nearly equal bacterial species diversity (11 against 10 in irrigated larval-positive sources and those without respectively). From the non-irrigated area, only one bacteria species was detected in habitats that lacked larvae (Fig 3).

## Bacterial isolates identified through sequencing irrigated and non-irrigated larval sources

Of the 31 sequenced isolates, 65% (20/31) were *Bacillus*, 13% (4/31) were *Escherichia*, 10% (3/31) were *Exiguobacterium*, and 3% (1/31) each were *Paenibacillus*, *Staphylococcus*, *Citrobacter*,

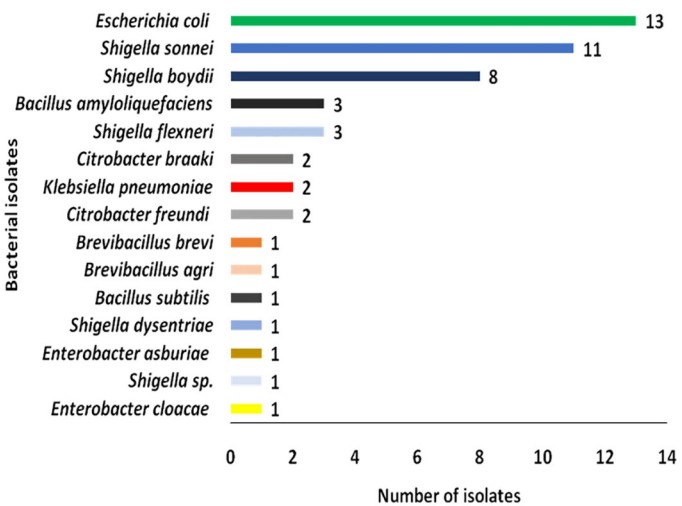

**Fig 2. Distribution of bacterial species detected by MALDI-TOF MS from irrigated and non-irrigated larval sources.**

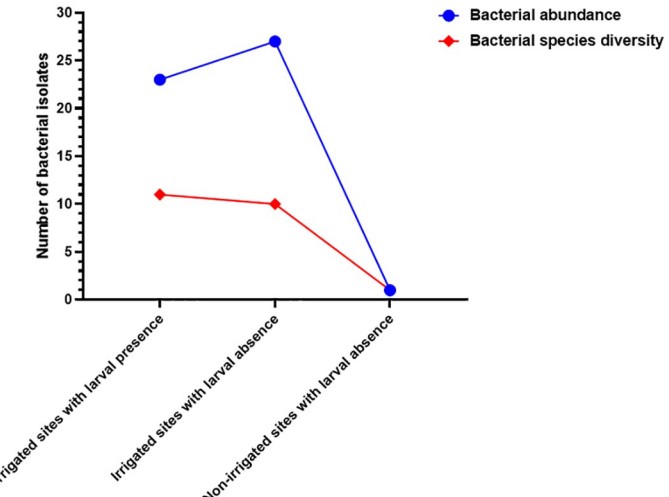

**Fig 3. Bacterial species diversity and abundance within aquatic habitats with larval presence or absence within irrigated and non-irrigated areas as detected by MALDI-TOF MS.**

and *Enterococcus* (Table 3). Among *Bacillus*, *B. velezensis* were the abundant species at a frequency of 25% (5/20), followed by *B. subtilis* at 20% (4/20), *B. siralis*, *B. stercoris*, *B. aerius*, and *B. cereus* each at 10% (2/20). The rest (*B. inaquosorum*, *B. mojavensis*, and *B. tequilensis*) were at a frequency of 5% (1/20) each (Table 3). The isolates clustered into 12 clades which were representative of the mentioned genera (Fig 4). *B. subtilis* (KC441741), *B. mojavensis* (OP218479 and OP482166), *B. tequilensis* (OP435764), and *B. stercoris* (OP521932) clustered with isolate NA10A, NA10B, NA12AA, NA4BB, and NA5AA in clade 1. *B. aerius* (OP115492) and isolates EMB6A and EMB6 were in clade 2. Clade 3 comprised of *Robertmurraya siralis* (OK570087), *B. siralis* (MH746088), and isolate NA4A; Clade 4 had *Staphylococcus arlettae* (OP402856) and isolate NA4A; Clade 5 had *E. gallinarum* (OP501807) and isolate NA9; Clade 6 had *B. cereus* (ON860698 and ON430535) and isolate NA3 and MC2; Clade 7 had *Exiguobacterium profundum* (OP793848 and OP263689) and isolates MC1A and NA8AA; Clade 8 had *E. coli* (CP102379), Uncultured bacterium (JQ265468) and isolate EMB9A; Clade 9 had *E. coli* (OP514801) and isolates NA8A, NA11A, MC5B, EMB9, and NASB; Clade 10 had *B. inaquosorium* (ON999044), *B. velezensis* (OP554433), *B. subtilis* (KF732994) isolates NA7C, NA2BA, NA5AB, NA6B, NA4BA, NA12B, and NA12AB; Clade 11 had *Citrobacter sp.* (OX245674) and

**Table 3. Frequency of isolated Genera from larval source in irrigated and non-irrigated areas based on sequencing and blasting results.**

| Genera | Irrigated area | | Non-irrigated area | |
|---|---|---|---|---|
| | Habitats with larval presence (n) | Habitats with larval absence (n) | Habitats with larval presence (n) | Habitats with larval absence (n) |
| *Bacillus* | 6 | 5 | 3 | 6 |
| *Citrobacter* | 0 | 1 | 0 | 0 |
| *Enterococcus* | 0 | 1 | 0 | 0 |
| *Escherichia* | 0 | 4 | 0 | 0 |
| *Exiguobacterium* | 3 | 0 | 0 | 0 |
| *Paenibacillus* | 0 | 1 | 0 | 0 |
| *Staphylococcus* | 0 | 0 | 1 | 0 |

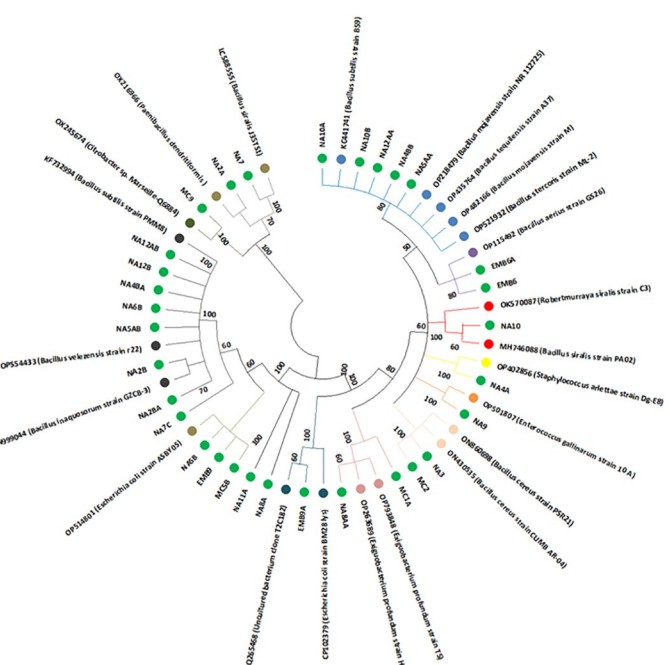

**Fig 4. Phylogenetic relationship between isolated bacteria and accessions within GenBank.**

isolate MC9 while Clade 12 comprised of *B. siralis* (LC588555), *Paenibacillus dendritiformis* (OX216966) and isolates NA2A, and NA7 (Fig 4).

The Maximum Likelihood method and the Tamura-Nei model were used to infer the evolutionary relationship of the 54 nucleotide sequences, 31 of which were local isolates and 23 were GenBank accessions. The tree with the highest log likelihood (-1502.82) is displayed. The percentage of trees in which related taxa clustered together is displayed next to the branches. Applying the Neighbor-Join and BioNJ algorithms to a matrix of pairwise distances calculated using the Tamura-Nei model, and then choosing the topology with the best log likelihood value, automatically constructed the initial tree(s) for the heuristic search. A discrete Gamma distribution was employed to explain variations in evolutionary rates between locations (5 categories (+G, parameter = 6.3041).

A Pearson's product-moment correlation was computed to assess the relationship between bacterial species abundance in larval sources in irrigated and non-irrigated areas and the presence or absence of irrigation. There was a strong, positive correlation between bacterial species abundance in larval sources in irrigated areas and the presence of irrigation, which was statistically significant (r = 0.334, n = 44, p = 0.027). Only gram-positive bacteria from the genera *Bacillus* and *Staphylococcus* were found in larval sources in non-irrigated area, with or without larvae. *Staphylococcus arlettae, Bacillus cereus, B. mojavensis, B. siralis, B. stercoris, B. subtilis,* and *B. velezensis* were among the species isolated. Except for *S. arlettae, B. stercoris,* and *B. velezensis* the other *Bacillus* species were found in non-larval-infested habitats. Only gram-positive bacteria, including *B. aerius, B. siralis, B. subtilis, B. tequilensis,* and *Exiguobacterium profundum*, were found in larval sources with larvae in irrigated areas. Potential larval sources with no larvae in irrigated areas had gram-negative (*Citrobacter sp., E. coli,* and Uncultured bacterium clone T2C182) and gram-positive bacteria (*B. cereus, B. inaquosorum, B. stercoris, B. subtilis, B. velezensis, P. dendritiformis,* and *E. gallinarum*).

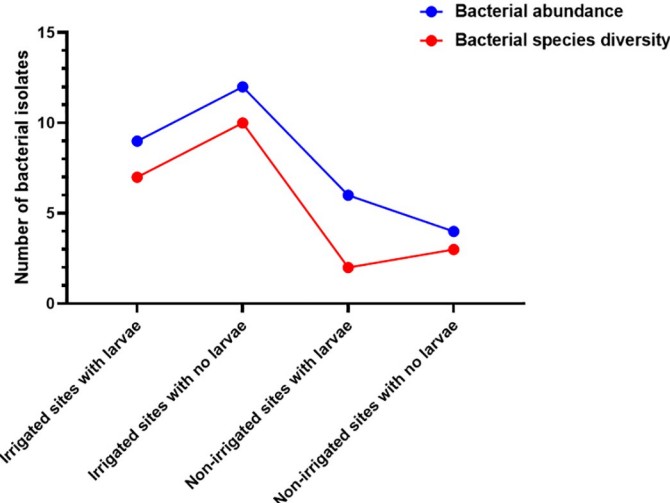

**Fig 5. Bacterial species diversity and abundance within habitats with larval presence or absence within irrigated and non-irrigated areas detected via sequencing.**

Generally, there was high bacterial abundance in each area as compared to diversity (Fig 5). Larval-positive habitats in irrigated and non-irrigated areas exhibited low bacterial diversity (7 and 2 respectively) as compared to larval-negative habitats in the same area (10 and 3, respectively). In terms of abundance, the habitats with larvae in irrigated and non-irrigated areas displayed varied patterns. The habitats in irrigated area had low abundance compared to habitats without larvae within the same area (9 against 12, respectively). However, in the non-irrigated area, aquatic habitats with larvae had high abundance than habitats with no larvae (6 against 4, respectively). Habitats with no larvae in irrigated areas had the highest bacterial abundance and diversity as compared to the rest followed closely by habitats within the same area with larvae (Fig 5).

## Metabolite analysis

Measurements were done for water samples with and without mosquito larvae from the irrigated and non-irrigated areas. These were each done in triplicate.

## Fatty acids concentration and larval availability

There were varying concentrations (0.95%-28.42%) of fatty acids (Palmitic, Palmitoleic, Stearic, Oleic, Linoleic, Linolenic, Arachidonic, Eicosapentaenoic acid (Epa), Docosahexaenoic acid (Dha), and Nervonic) in larval habitats within irrigated and non-irrigated areas with larval presence or absence (Table 4). Fatty acid concentrations in habitats with larvae within irrigated and non-irrigated areas, and habitats with no larvae in irrigated and non-irrigated areas did not differ significantly, implying that they had no overall effect on larval presence or absence ($F$ (3, 36) = 6.732e-008, P>0.99). At the fatty acid component level, larval presence was observed in irrigated and non-irrigated areas where Stearic concentrations were low (5.6% and 6.3% in irrigated and non-irrigated areas, respectively). Contrary to what was observed at low Stearic concentrations, larval presence was observed at elevated Nervonic concentrations of 6% and 6.2% in irrigated and non-irrigated areas, respectively (Fig 6).

**Table 4. Fatty acid concentrations in larval habitats with and with no larvae within irrigated and non-irrigated areas.**

| Fatty acids | Fatty acid concentration (%) in habitats in the irrigated area | | Fatty acid concentration (%) in habitats in the non-irrigated area | |
| --- | --- | --- | --- | --- |
| | Larval positive habitats | Larval negative habitats | Larval positive habitats | Larval negative habitats |
| Palmitic | 28.42 | 28.28 | 13.77 | 22.45 |
| Palmitoleic | 16.49 | 8.74 | 12.39 | 12.36 |
| Stearic | 5.60 | 10.76 | 6.29 | 14.63 |
| Oleic | 14.68 | 11.90 | 12.28 | 26.07 |
| Linoleic | 7.07 | 4.10 | 9.31 | 5.33 |
| Linolenic | 8.31 | 3.40 | 6.86 | 7.28 |
| Arachidonic | 5.99 | 10.45 | 10.01 | 4.92 |
| Epa | 3.98 | 5.27 | 10.65 | 2.19 |
| Dha | 3.44 | 13.38 | 12.22 | 3.82 |
| Nervonic | 6.01 | 3.72 | 6.21 | 0.95 |
| **Average fatty acid** | 10 | 10 | 10 | 10 |

## Phytochemical concentration and larval availability

In both larval breeding sites in irrigated and non-irrigated areas, the crude fat content ranged from 0.01% to 0.18%. Larval-positive habitats in the irrigated area had a lower crude fat content of 0.01% than the same site without larvae (0.18%). The crude fat content (0.01%) was the same in non-irrigated area with and without larvae. Larval habitats within irrigated areas also recorded high crude protein content (0.13% and 0.16% in larval habitats within irrigated areas with and with no larvae, respectively) compared to those in non-irrigated areas (0.08% and 0.07%). Larvae were found in irrigated areas with low crude protein content (0.13% against 0.16%) as opposed to what was observed in non-irrigated areas (0.08% against 0.07%). Similar observation was made on Beta-Carotene (μg/100g) content in irrigated areas (1.26 against 2.04) and non-irrigated areas (5.2 against 3.16) as well as total phenol μg (GAE)/100g (Table 5). Larval presence was evident in aquatic habitats within irrigated and non-irrigated areas with high total chlorophyll content (1.12 μg/g against 0.81μg/g and 3.37 μg/g against 0.82 μg/g). Both Tannin and anti-oxidant contents were low in habitats with larvae in irrigated areas and high in habitats with larvae in non-irrigated areas (Table 5). The varied

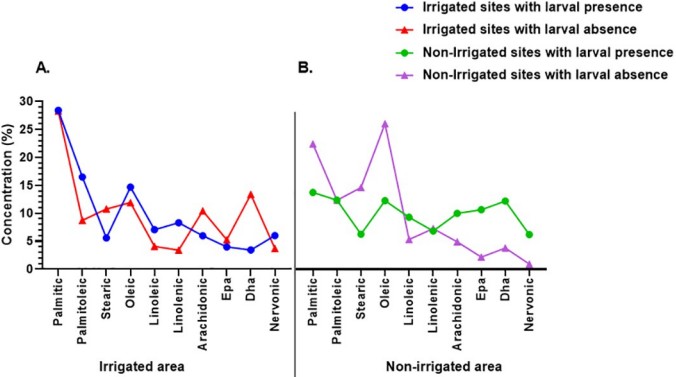

**Fig 6. Component of fatty acid concentration in habitats with larval presence or absence within irrigated and non-irrigated areas.** The concentration of each fatty acid component in larval habitats in an irrigated area (**A**). The concentration of each fatty acid component in larval habitats in non-irrigated area (**B**).

**Table 5. Phytochemical concentrations in larval habitats with and with no larvae within irrigated and non-irrigated areas.**

| Element | Irrigated area | | Non-irrigated area | |
|---|---|---|---|---|
| | Larval positive habitats | Larval negative habitats | Larval positive habitats | Larval negative habitats |
| Crude Fat (%) | 0.04 | 0.18 | 0.01 | 0.01 |
| Crude Protein (%) | 0.13 | 0.16 | 0.08 | 0.07 |
| Beta-Carotene (µg/100g) | 1.26 | 2.04 | 5.20 | 3.16 |
| Total Phenol (µg (GAE)/100g) | 4.49 | 5.29 | 4.55 | 4.68 |
| Total Chlorophyll (µg/g) | 1.12 | 0.81 | 3.37 | 0.82 |
| Tannin (mg (TAE)/100g) | 0.10 | 1.08 | 3.22 | 0.28 |
| Anti-oxidant (@0.5ml/ml) % Inhibition | 20.38 | 29.40 | 26.40 | 23.65 |

concentration of each phytochemical component between habitats with larvae and with no larvae in irrigated and non-irrigated areas was not significant.

## Sugar concentration and larval availability

Habitats with larvae in irrigated areas had high concentrations (mg/100ml) of Pectin (0.64 against 0.50), cellulose (7.72 against 2.86), and fructose (0.36 against 0.33) as compared to habitats with no larvae. Similar trend on pectin, cellulose, and fructose was observed in habitats with and without larvae in irrigated areas (Table 6). The three sugar components were at higher concentration in larval habitats within non-irrigated areas with larvae as compared to habitats with larvae in irrigated areas (Fig 7). Glucose concentration was however low in habitats with larvae in irrigated areas compared to habitats with no larvae (0.38 against 1.16). In non-irrigated areas, glucose concentration was high in habitats with larvae as compared to those without (2.67 against 0.47). Similar observation was made on sucrose concentration in irrigated (0.08 against 0.16) and non-irrigated areas (0.25 against 0.09). The varied concentration of each sugar component between habitats with larvae and with no larvae in irrigated and non-irrigated areas was not significant.

## Mineral concentration and larval availability

High concentrations of manganese, calcium, and copper were observed in habitats with larvae in irrigated and non-irrigated areas as compared to those without (Table 7). Other minerals such as lead, iron, magnesium, and zinc were at near similar concentrations in all habitats (Fig 8).

## Discussion

This study revealed important aspects that can be used in the development of vector control tools. *Shigella* and *Bacillus* genera were observed to be dominant in habitats in the irrigated

**Table 6. Sugar concentrations in larval habitats with and with no larvae within irrigated and non-irrigated areas.**

| Components | Irrigated area | | Non-irrigated area | |
|---|---|---|---|---|
| | Larval positive habitats | Larval negative habitats | Larval positive habitats | Larval negative habitats |
| Pectin (mg/100ml) | 0.64 | 0.50 | 0.77 | 0.19 |
| Cellulose (mg/100ml) | 7.72 | 2.86 | 11.17 | 0.86 |
| Glucose (mg/100ml) | 0.38 | 1.16 | 2.67 | 0.47 |
| Fructose (mg/100ml) | 0.36 | 0.33 | 0.76 | 0.25 |
| Sucrose (mg/100ml) | 0.08 | 0.16 | 0.25 | 0.09 |

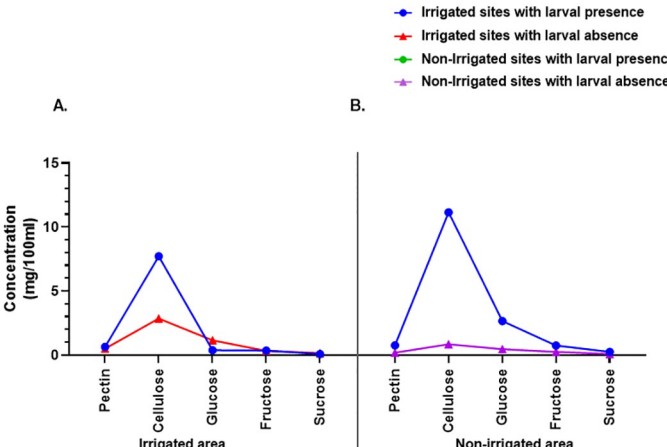

**Fig 7. Component of sugar concentration in habitats with larval presence or absence within irrigated and non-irrigated areas.** The concentration of each sugar component in larval habitats in an irrigated area (**A**). The concentration of each sugar component in larval habitats in non-irrigated area (**B**).

and non-irrigated areas respectively. Irrigation positively influenced bacterial abundance and diversity. Larvicidal isolates *Brevibacillus brevis*, *Bacillus subtilis*, and *Exiguobacterium profundum* were isolated and grouped with *Bacillus mojavensis*, *Bacillus tequilensis*, *Bacillus stercoris*, and *Brevibacillus agri*. Less proteins and fats were detected in the irrigated area from both larval-positive and larval larval-negative habitats as compared to the non-irrigated area, while high chlorophyll and sugar content was associated with larval presence in both areas. In addition, substantial concentrations of Manganese, Calcium, and Copper were found in sites containing larvae in both irrigated and non-irrigated areas.

*Bacillus amyloliquefaciens*, a subspecies of the *B. subtilis* is known to be a bio-control against plant pathogens [48–50]. This species has also been observed to reduce harmful water components that reduce water oxygen content like ammonia, total nitrogen, total phosphorus, and chemical oxygen demand while increasing nitrates [51]. The presence of this species as the only bacteria identified via MALDI-TOF in the non-irrigated area confirms that *B. amyloliquefaciens* significantly affects the bacterial community diversity and composition in the non-irrigated area. This is similar to studies done by Yang et al., [51]. The presence of *B. amyloliquefaciens* in the long-standing aquatic habitats in the non-irrigated area confirms that the species does not colonize a native bacterial community in a freshwater aquatic environment [51, 52].

**Table 7. Mineral concentrations in larval habitats with and with no larvae within irrigated and non-irrigated areas.**

| Minerals | Irrigated area | | Non-irrigated area | |
|---|---|---|---|---|
| | Larval positive habitats | Larval negative habitats | Larval positive habitats | Larval negative habitats |
| Lead (mg/100ml) | 5.70 | 5.41 | 8.62 | 9.47 |
| Manganese (mg/100ml) | 249.72 | 153.83 | 198.04 | 182.48 |
| Calcium (mg/100ml) | 8.46 | 7.89 | 9.10 | 6.51 |
| Copper (mg/100ml) | 4.97 | 3.66 | 6.25 | 5.00 |
| Iron (mg/100ml) | 1.16 | 1.08 | 1.11 | 1.16 |
| Magnesium (mg/100ml) | 1.12 | 1.12 | 1.16 | 1.12 |
| Zinc (mg/100ml) | 0.02 | 0.00 | 0.01 | 0.01 |

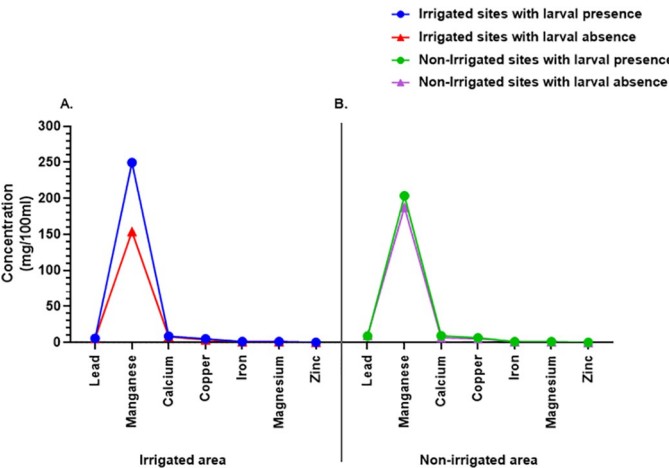

**Fig 8. Mineral concentration in habitats with larval presence or absence within irrigated and non-irrigated areas.**
The concentration of minerals in larval habitats in an irrigated area (**A**). The concentration of minerals in larval habitats in non-irrigated area (**B**).

Based on MALDI-TOF MS and sequencing results, larval sources in irrigated areas had higher bacterial species diversity and abundance than sources in non-irrigated areas. Similar to previous research that linked increased soil bacterial abundance and diversity to irrigation [53, 54], the findings of this study confirm the influence of irrigation on increased bacterial abundance and species diversity that may influence larval productivity on larval sources or pools. MALDI-TOF MS, revealed *Shigella* and *Escherichia* as the dominant genera in larval sources with larvae presence in irrigated areas. Except for *Brevibacillus*, some species of the former two and other isolated genera (*Bacillus*, *Klebsiella*, *Citrobacter*, and *Enterobacter*) are considered endosymbionts and have been isolated from the gut of *Anopheles* [55–57]. Except for *Bacillus sphaericus*, which reduces vectoral capacity [58], species from the five genera have a significant influence on nutrient assimilation, vectoral capacity, and membrane formation in mosquitoes, explaining the importance of their presence in Homa Bay larval sources [59–61]. Members of the genera have also been linked to the synthesis of kairomones that attract and stimulate oviposition in vectors such as *Aedes aegypti* and *Ae. Albopictus* [62].

The exempted genus *Brevibacillus* has previously been linked to entomopathogenic activities, with the spores produced by *Brevibacillus laterosporus* and *Brevibacillus brevis* being highly toxic to both larvae and adult mosquitoes [63, 64]. The presence of *Brevibacillus brevis* may have an effect on larval densities in larval sources with larvae in irrigated areas in the study site, interfering with their overall larval productivity. Furthermore, because *Brevibacillus agri* is closely related to *Brevibacillus brevis*, it should be tested for entomopathogenic characteristics [65]. This will determine whether the bacteria play a role in larval reduction or forms bio-larvicidal spores and can be used in larval source management (LSM).

Additional four bacterial taxa (*Enterococcus*, *Exiguobacterium*, *Paenibacillus*, and *Staphylococcus*) were found in larval sources in irrigated and non-irrigated parts of Homa Bay, according to sequencing data. Among the key identified bacteria that were only present in larval present sources in irrigated areas of Homa Bay was *Exiguobacterium profundum*. The bacterium is known to impair mosquito fecundity, egg hatchability, and larval development, and it may have a detrimental impact on larval productivity in larval sources inside

irrigated regions [66]. Although present in potential larval habitats that lacked *Anopheles* larvae, *Bacillus subtilis* is renowned for generating bio-surfactant surfactin, which is a bio-adulticide to *An. stephensi* [67], as well as other secondary metabolites that are bio-larvicides to *A. aegypti* [68]. The finding of clustering or grouping with other isolated bacteria such as *B. mojavensis*, *B. tequilensis*, and *B. stercoris* (found in larvae-infested larval sources) suggests that the larvicidal or adulticidal capability of these bacteria should be investigated.

Stearic and oleic acids are fatty acids with long hydro-carbon chains. The absence of larvae in aquatic habitats with high levels of stearic acid could be as a result of the nature of this fatty acid. Stearic acid is known to be waxy and slightly soluble in water thus forming a thin layer on water surfaces [69]. This layer restricts the entry of oxygen into the water. This could be a hindrance to oviposition adult mosquitoes as they tend to prefer habitats that will support the survival of the immature. Stearic acid has been observed to be potent to mosquito larvae [70]. Oleic acid, a monounsaturated fatty acid, soluble in water, is known to be beneficial to other living organisms. However oleic and linoleic acids have been observed to be lethal to mosquito larvae in high concentrations [70, 71]. This could be the reason for the optimal concentrations of these fatty acids in habitats with larvae. Linoleic acid has been observed to be beneficial for mosquito adult development [72].

A high concentration of total chlorophyll was observed to support habitat infestation by mosquito larvae. This could be due to the presence of algae matter which acts as food for the immature to support their growth. This is consistent with other studies that observed that algae was used as food for mosquito larvae [73–76].

The nutritional capacity of mosquito larvae is crucial for the growth of the larvae and the emergence of adult mosquitoes. Inadequately nourished larvae may fail to develop or adversely affect the size and reproductive capacity of the resultant adult mosquitoes. Generally, high concentrations of simple sugar were associated with larval presence in the habitats. This was observed for pectin, cellulose, glucose, fructose, and sucrose. These are known as soluble sugars and ready food products that are used by mosquito larvae for survival.

Similarly, manganese was found at higher concentrations than other metals throughout larval habitats in irrigated and non-irrigated regions with larval presence. Manganese, in conjunction with calcium, chlorine, iron, potassium, magnesium, sodium, sulfur, and phosphorus, is known to promote normal mosquito larval development [77, 78]. The majority of the other metals, including lead, calcium, copper, iron, magnesium, and zinc, were found in extremely low concentrations. Although copper is required for enzyme functioning in larval pigmentation, oxidative stress protection, and respiration, excessive concentrations are known to be larvicidal [79, 80], and has also been linked to mosquito resistance to insecticides such as lambda-cyhalothrin [81].

## Conclusion

Irrigation increased bacterial species abundance and diversity in larval sources within the irrigated area. *Shigella* and *Escherichia* species were the common genera in larval sources with larvae presence in irrigated areas. In the irrigated region, notable larvicidal bacteria *B. brevis*, *B. subtilis*, and *E. profundum* were found. Moreover, the larvicidal or adulticidal properties of *B. mojavensis*, *B. tequilensis*, *B. stercoris*, and *Brevibacillus agri* should be examined since they clustered with known entomopathogenic bacteria in this study. Finally, high fatty acid concentration may have an impact on larval availability in both irrigated and non-irrigated areas, in addition to chlorophyll, sugar, and manganese promoted larval availability in the aquatic habitats.

## Supporting information

**S1 File.**
(DOCX)

**S1 Table.**
(DOCX)

## Acknowledgments

The authors wish to thank the community and the leaders of Homa Bay County for allowing us to work in the area. We also acknowledge the sub-Saharan Africa ICEMR field team and staff for providing professional and technical support during the study; more specifically Sally Mongoi, Fredrick Odongo, and Thomas Katiye. In addition, we would like to thank Anne Owiti, Sebastian Musundi, and David Abuga for their technical support provided.

## Author Contributions

**Conceptualization:** Pauline Winnie Orondo, Andrew K. Githeko, Steven G. Nyanjom.

**Data curation:** Pauline Winnie Orondo.

**Formal analysis:** Pauline Winnie Orondo, Kevin O. Ochwedo.

**Funding acquisition:** Pauline Winnie Orondo, Steven G. Nyanjom.

**Investigation:** Pauline Winnie Orondo, Steven G. Nyanjom.

**Methodology:** Pauline Winnie Orondo, Guiyun Yan, Andrew K. Githeko, Steven G. Nyanjom.

**Project administration:** Pauline Winnie Orondo, Harrysone Atieli, Andrew K. Githeko, Steven G. Nyanjom.

**Resources:** Pauline Winnie Orondo, Guiyun Yan, Steven G. Nyanjom.

**Software:** Pauline Winnie Orondo.

**Supervision:** Pauline Winnie Orondo, Harrysone Atieli, Andrew K. Githeko, Steven G. Nyanjom.

**Validation:** Pauline Winnie Orondo, Kevin O. Ochwedo.

**Visualization:** Pauline Winnie Orondo.

**Writing – original draft:** Pauline Winnie Orondo, Kevin O. Ochwedo, Steven G. Nyanjom.

**Writing – review & editing:** Pauline Winnie Orondo, Kevin O. Ochwedo, Harrysone Atieli, Guiyun Yan, Andrew K. Githeko, Steven G. Nyanjom.

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
