## [Decision Letter · Decision Letter 0]

3 May 2023

PONE-D-23-05982Effects of bacterial composition and aquatic habitat metabolites on malaria vector larval availability in irrigated and non- irrigated sites of Homa Bay county, western KenyaPLOS ONE

Dear Dr. Orondo,

Thank you for submitting your manuscript to PLOS ONE. After careful consideration, we feel that it has merit but does not fully meet PLOS ONE’s publication criteria as it currently stands. Therefore, we invite you to submit a revised version of the manuscript that addresses the points raised during the review process.

 Please, carfeully consider the suggestions done by the reviewers

We look forward to receiving your revised manuscript.

Kind regards,

Guadalupe Virginia Nevárez-Moorillón, Ph.D.

Academic Editor

PLOS ONE

Journal Requirements:

3. Thank you for submitting the above manuscript to PLOS ONE. During our internal evaluation of the manuscript, we found significant text overlap between your submission and previous work in the [introduction, conclusion, etc.].

Please revise the manuscript to rephrase the duplicated text, cite your sources, and provide details as to how the current manuscript advances on previous work. Please note that further consideration is dependent on the submission of a manuscript that addresses these concerns about the overlap in text with published work.

[If the overlap is with the authors’ own works: Moreover, upon submission, authors must confirm that the manuscript, or any related manuscript, is not currently under consideration or accepted elsewhere. If related work has been submitted to PLOS ONE or elsewhere, authors must include a copy with the submitted article. Reviewers will be asked to comment on the overlap between related submissions (http://journals.plos.org/plosone/s/submission-guidelines#loc-related-manuscripts).]

We will carefully review your manuscript upon resubmission and further consideration of the manuscript is dependent on the text overlap being addressed in full. Please ensure that your revision is thorough as failure to address the concerns to our satisfaction may result in your submission not being considered further

4. We note that you have stated that you will provide repository information for your data at acceptance. Should your manuscript be accepted for publication, we will hold it until you provide the relevant accession numbers or DOIs necessary to access your data. If you wish to make changes to your Data Availability statement, please describe these changes in your cover letter and we will update your Data Availability statement to reflect the information you provide

Reviewers' comments:

Reviewer's Responses to Questions

**Comments to the Author**

1. Is the manuscript technically sound, and do the data support the conclusions?

Reviewer #1: Yes

Reviewer #2: Yes

2. Has the statistical analysis been performed appropriately and rigorously? 

Reviewer #1: Yes

Reviewer #2: No

3. Have the authors made all data underlying the findings in their manuscript fully available?

Reviewer #1: Yes

Reviewer #2: Yes

4. Is the manuscript presented in an intelligible fashion and written in standard English?

Reviewer #1: Yes

Reviewer #2: Yes

5. Review Comments to the Author

Reviewer #1: I am pleased to review miss Pauline and others entitled "Effects of bacterial composition and aquatic habitat metabolites on malaria vector larval availability in irrigated and non-irrigated sites of Homa Bay county, western Kenya". This manuscript has taken in the moment when larval source management has become the core intervention by WHO to answer some questions about habitats in irrigation and non-irrigation(rain-fed areas). This paper has been written in inteligently manner and all sections have been connected;

I propose the paper to be ACCEPTED AFTER THE MINOR REVISION i HAVE MADE IN THE ATTACHED MS.

1. The citation should be changed to square brackets

2. take care of my inputs in two parts in the attached document.

3. Check the reference list to be clear to the journal format.

If these three are done, kindly accept the manuscript.

Reviewer #2: Focus of the study: Overall, the bacterial composition and aquatic habitat metabolites can have complex and varied effects on the availability of malaria vector larvae in both irrigated and non-irrigated sites. Understanding these factors is important for developing effective strategies to control the spread of malaria. However, I have few major observations that should be clarified

1. Is the bacterial composition and aquatic habitat metabolites in the aquatic habitats affected by seasonality?

2. Did you assess other factors such as aquatic predators?

3. In the irrigation and non-irrigation, what was the major crop? Did you recorded any use of pesticides? That could have impacts to mosquito larvae and water quality?

4. Sample collection: The water samples were pooled into 12 bottle sets and used for microbial analysis. Why 12 bottles? What was the time interval between sample collection and analysis? Was there any preservatives added in water samples?

5. Major observation: The article focus mainly on water sample analysis, the author need to explain how the mosquito larvae were collected, identified, which keys and how were they correlated with bacterial composition and aquatic habitat metabolites

6. My suggestion would be add a section describing how mosquito larvae were collected and processed as well. The analysis section should also show how was the analysis of mosquitoes done and how each factor observed influenced abundance/presence of mosquito larvae

7. Map of study area showing irrigated and non-irrigated sites?

8. Analysis section should show how these metabolites affect abundance of mosquito larvae

9. Conclusion: “In addition, high fatty acid concentration may have an impact on larval production in both irrigated and non-irrigated areas whereas chlorophyll, sugar, and manganese promoted growth” How did you come up with this conclusion? Please refer to the comment No. 8 above. These should be analysis to assess how this affect larval abundance. Off course a GLMM approach can be used to count for random and fixed variables

6. PLOS authors have the option to publish the peer review history of their article (what does this mean?). If published, this will include your full peer review and any attached files.

Reviewer #1: **Yes: **Prof. Eliningaya Kweka

Reviewer #2: No

---

## [Author Response · Author response to Decision Letter 0]

16 May 2023

Reviewer 1

1) The citation should be changed to square brackets 

This has done throughout the document

2) Take care of my inputs in two parts in the attached document. 

Amendments done (line 103-104 and 460)

3) Check the reference list to be clear to the journal format. 

Ammended as suggested

Reviewer 2

1) Is the bacterial composition and aquatic habitat metabolites in the aquatic habitats affected by seasonality? 

Yes, seasonality has been observed to affect bacterial and metabolite composition in aquatic habitats as has been demonstrated by previous studies (Yi et al., 2021). This study was done from Nov-Jan which is generally the end of short rains and the beginning of the dry season in Kenya. Financial constraints could however not allow for more data to be collected to analyse the effects of seasonality. Further studies can be conducted to assess this effect in the irrigated and non- irrigated area

2) Did you assess other factors such as aquatic predators? 

Yes, larval and predator densities were also assessed, however, the interest of this research was the microbial and metabolite composition and how they affect larval presence/ absence. Other confounding factors were not analyzed in this study

3) In the irrigation and non-irrigation, what was the major crop? Did you recorded any use of pesticides? That could have impacts to mosquito larvae and water quality? 

Agricultural activities have been indicated in Line 104-108

Information on pesticide use in the region is already published in previous research (Orondo et al., 2021) 

4) Sample collection:

The water samples were pooled into 12 bottle sets and used for microbial analysis. Why 12 bottles? What was the time interval between sample collection and analysis? Was there any preservatives added in water samples? 

12 sets were used. Pools were done every month from Nov 2021 to Jan 2022 (Line112) in both irrigated and non- irrigated area and larval positive and negative habitats. This resulted in 4 pools monthly. Below is a diagrammatic illustration of water sample collection each month 

All analysis were done between January- February 2022. The (line 118) indicates how the water samples were stored and preserved awaiting analysis

5) Major observation:

The article focus mainly on water sample analysis, the author need to explain how the mosquito larvae were collected, identified, which keys and how were they correlated with bacterial composition and aquatic habitat metabolites

A statement on larval sampling ahs been included in Line 119-121. However, the main focus of this study was larval availability in relation to water and metabolite composition and that is why mosquito densities were not included

6) My suggestion would be add a section describing how mosquito larvae were collected and processed as well. The analysis section should also show how was the analysis of mosquitoes done and how each factor observed influenced abundance/presence of mosquito larvae 

Line 119-121 describes larval sampling and morphological identification.

As the main focus of this study was larval availability in relation to water and metabolite composition, further analysis on larvae were not included

7) Map of study area showing irrigated and non-irrigated sites? 

A map has been included (Fig1)

8) Analysis section should show how these metabolites affect abundance of mosquito larvae 

The study focus (as indicated in the title) was on larval presence/ availability and not larval abundance and that is why mosquito densities were not included

9) Conclusion:

“In addition, high fatty acid concentration may have an impact on larval production in both irrigated and non-irrigated areas whereas chlorophyll, sugar, and manganese promoted growth” How did you come up with this conclusion? Please refer to the comment No. 8 above. These should be analysis to assess how this affect larval abundance. Off course a GLMM approach can be used to count for random and fixed variables 

This statement (Line 534-537) has been re-phrased to “Finally, high fatty acid concentration may have an impact on larval availability in both irrigated and non-irrigated areas, in addition to chlorophyll, sugar, and manganese promoted larval availability in the aquatic habitats”.

We believe that this is a clearer statement to explain what the study was looking into.

---

## [Editor Report · Decision Letter 1]

17 May 2023

Effects of bacterial composition and aquatic habitat metabolites on malaria vector larval availability in irrigated and non-irrigated sites of Homa Bay county, western Kenya

PONE-D-23-05982R1

Dear Dr. Orondo,

We’re pleased to inform you that your manuscript has been judged scientifically suitable for publication and will be formally accepted for publication once it meets all outstanding technical requirements.

Kind regards,

Guadalupe Virginia Nevárez-Moorillón, Ph.D.

Academic Editor

PLOS ONE
---

## [Editor Report · Acceptance letter]

24 May 2023

PONE-D-23-05982R1 

Effects of bacterial composition and aquatic habitat metabolites on malaria vector larval availability in irrigated and non-irrigated sites of Homa Bay county, western Kenya 

Dear Dr. Orondo:

I'm pleased to inform you that your manuscript has been deemed suitable for publication in PLOS ONE. Congratulations! Your manuscript is now with our production department. 

Kind regards, 

on behalf of

Dr. Guadalupe Virginia Nevárez-Moorillón 

Academic Editor

PLOS ONE